# Recognition and Depth Estimation of Ships Based on Binocular Stereo Vision

Yuanzhou Zheng [1,2], Peng Liu [1,2], Long Qian [1,2,*], Shiquan Qin [1,2], Xinyu Liu [1,2], Yong Ma [1,2] and Ganjun Cheng [3]

1   School of Navigation, Wuhan University of Technology, Wuhan 430036, China
2   Hubei Key Laboratory of Inland Shipping Technology, Wuhan 430036, China
3   Yangtze River Navigation Administration, Ministry of Transport, Wuhan 430036, China
*   Correspondence: lqian@whut.edu.cn; Tel.: +86-15-(62)-3856816

**Abstract:** To improve the navigation safety of inland river ships and enrich the methods of environmental perception, this paper studies the recognition and depth estimation of inland river ships based on binocular stereo vision (BSV). In the stage of ship recognition, considering the computational pressure brought by the huge network parameters of the classic YOLOv4 model, the MobileNetV1 network was proposed as the feature extraction module of the YOLOv4 model. The results indicate that the mAP value of the MobileNetV1-YOLOv4 model reaches 89.25%, the weight size of the backbone network was only 47.6 M, which greatly reduced the amount of computation while ensuring the recognition accuracy. In the stage of depth estimation, this paper proposes a feature point detection and matching algorithm based on the ORB algorithm at sub-pixel level, that is, firstly, the FSRCNN algorithm was used to perform super-resolution reconstruction of the original image, to further increase the density of image feature points and detection accuracy, which was more conducive to the calculation of the image parallax value. The ships' depth estimation results indicate that when the distance to the target is about 300 m, the depth estimation error is less than 3%, which meets the depth estimation needs of inland ships. The ship target recognition and depth estimation technology based on BSV proposed in this paper makes up for the shortcomings of the existing environmental perception methods, improves the navigation safety of ships to a certain extent, and greatly promotes the development of intelligent ships in the future.

**Keywords:** navigation safety; environmental perception; binocular stereo vision; MobileNetV1-YOLOv4; FSRCNN; ORB

## 1. Introduction

With the deepening of economic globalization in the 21st century, water transportation, as an important mode of transportation; it carries nearly 90% of the freight volume of global bulk trade, which greatly promotes the development of the world and regional economies [1]. At present, our country's shipping industry has also made significant progress, and the number of marine (especially inland river) ships is increasing yearly, which not only promotes the national economic development, but also leads to the rising trend of ship traffic accidents. Therefore, it is a key issue to improve the safety of ship navigation in ports or waters with high and complex traffic density [2,3]. Vessel Traffic Service (VTS) [4] can effectively supervise the navigation situation of ships and reduce the occurrence of marine traffic accidents to a certain extent. However, the inland river environment is complex and changeable, and the density of ships in the waterway is high, which greatly limits the role of VTS. The key to the safe navigation of ships lies in the effective perception of the surrounding navigation environment, and the drivers can make timely and correct decisions based on the obtained information. Ship Automatic Identification System (AIS) [5] is currently the primary environmental perception means in the shipping industry. However, because some ships do not keep the AIS in normal working state as required, and do not enter the accurate information of the ship in the

equipment according to the regulations, this affects the navigation safety of ships to a certain extent. Therefore, how to make up for the deficiency of the existing perception means and further improve the maritime supervision capability has become an important issue in the field of ship safety.

At present, intelligent shipping [6,7] has gradually become a new development trend. Countries around the world are conducting research on the direction of intelligent water transportation, using various sensors, communication equipment, and other methods to intelligently perceive and receive information about ship data, navigation environment, ports, and wharves. Then big data, machine learning, image processing, pattern recognition and other methods are adopted for effective information processing to conduct analysis, evaluation and decision-making, and to improve the safety of ship navigation [8,9]. As an interdisciplinary subject, computer vision (CV) [10] further promoted the development of intelligent shipping.

As a branch of CV, binocular stereo vision (BSV) [11] technology is gradually becoming mature, which is mainly composed of binocular camera calibration [12], image stereo matching [13], and depth calculation. By simulating human eyes, the two cameras shoot the same target scene at the same time and directly process the surrounding environment information, and then realize the recognition and distance measurement of the target in the three-dimensional scene by the principle of triangulation [14]. The technology has a simple structure, high flexibility and reliability, and is widely used in 3D image reconstruction [15], robot navigation [16], and quality inspection of industrial products [17]. The University of Washington and Microsoft installed BSV technology on the Mars Reconnaissance Orbiter [11]. By taking images of celestial bodies at different positions and using coordinate transformation to restore the three-dimensional coordinates of space points, a relatively accurate celestial landscape can be obtained. Ma [18] combined UAV with BSV perception technology to build an automatic detection system for transmission lines, which realized automatic real-time detection of transmission line components with good robustness and accuracy. Zhang [19] established a target distance measurement model based on deep learning and BSV, obtained the internal and external parameters of the camera through calibration, used the Faster R-CNN algorithm to identify the target, and brought the obtained feature points into the BSV model, then obtained the depth information of the target object. However, this model only realizes the recognition and ranging of a single target, and the applicability is weak. Ding [20] proposed a reliable and stable moving target localization method based on BSV. First, the O-DHS algorithm was used to separate the recognized target from the complex background, and the S-DM algorithm was used to perform depth analysis on the extracted feature points, finally calculated the distance and three-dimensional information through coordinate transformation. This method has high positioning accuracy and recognition accuracy, but the algorithm takes a long time to execute, and the effect is unsatisfactory in complex environments. Based on the principle of BSV, Li [21] established a non-contact displacement measurement system, which uses template matching to extract the image coordinates of the measurement points, and recovers its spatial position information through Euclidean 3D reconstruction. This greatly improved the practicability of the measurement system, but the algorithm is sensitive to the size of the template matching, which affects the calculation of the position information to a certain extent. Liu [22] focused on the camera calibration of BSV and proposed an online calibration method based on dual parallel cylindrical targets and linear laser projectors. This method does not require feature points or contour information of the target image, and the camera calibration process can be completed by obtaining the laser strip of the target image, but the algorithm has a high computational cost and the universality is weak. Reference [23] proposed a high-speed stereo matching algorithm suitable for ultra-high-resolution binocular images, which uses small-sized images to match close-range targets, and large-sized images to match distant targets. The disparity map of left and right images was obtained by combining the results of hierarchical matching, the matching cost is calculated by the image pyramid strategy, which greatly shortens

the time consumption in the matching process, however its matching accuracy must be further improved.

In order to further improve the safety of ship navigation and overcome the shortcomings of the existing navigation environment perception methods, based on the above research, this paper applies BSV technology to the recognition and depth estimation of inland ships. By loading the ships with "eyes", the technology perceives the navigation environment with visual information, and enables the ships to make timely and effective decisions through information interaction, so as to the purpose of safe navigation of ships. This work is primarily divided into two stages: ship recognition and depth estimation. In the ship target recognition stage, based on the classic YOLOv4 network model, considering the computational pressure brought by the huge network parameters of the model, a lightweight network is proposed to complete the recognition task, that is, the MobileNetV1 network is used to replace the feature extraction network CSPDarknet53 of the YOLOv4 model. Then this paper establishes the MobileNetV1-YOLOv4 ship target recognition model which greatly reduces the amount of computation while ensuring the recognition accuracy. In the stage of depth estimation, the BSV depth estimation model is first established, and then the FSRCNN network is used to reconstruct the original image pairs with super-resolution to further enhance the ship feature information, then the ORB algorithm is adopted to detect ship feature at the sub-pixel level, the parallax value between the image pairs is obtained by stereo feature match, finally the depth information of the ship is obtained through the principle of triangulation and coordinate transformation.

## 2. BSV Depth Estimation Model

The BSV depth estimation technology perceives the depth of the surrounding environment through an anthropomorphic method, and obtains the three-dimensional information of the target in the real world. According to the principle of triangulation, two parallel and coplanar cameras are used to capture the same scene from different angles, and the depth information is recovered by calculating the parallax value between the image pairs. As shown in Figure 1, the optical center positions of the left and right cameras are $O_l$ and $O_r$ respectively; $O_l - X_l Y_l Z_l$ and $O_r - X_r Y_r Z_r$ are the left and right camera coordinate systems; $b$ is the horizontal distance between the optical centers, called the baseline distance; The focal length of the camera is $f$; For the three-dimensional space point $P(X, Y, Z)$, its projection point coordinates in the imaging coordinate system of the left and right cameras are $p(x_l, y_l)$ and $p(x_r, y_r)$ respectively.

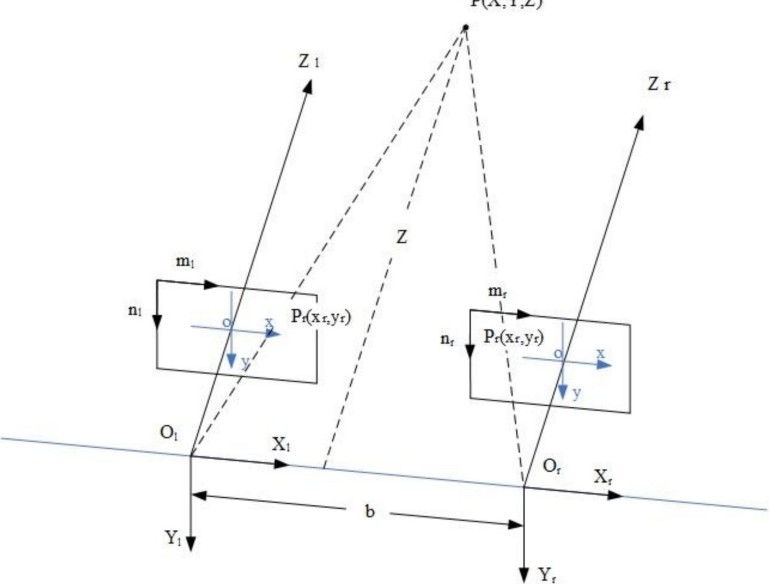

**Figure 1.** The stereo model of BSV.

Project the above stereo model to the *XOZ* plane, as shown in Figure 2.

**Figure 2.** Projection plane of BSV model.

According to the triangle similarity principle:

$$\begin{cases} \frac{z}{f} = \frac{x}{x_l} \\ \frac{z}{f} = \frac{x-b}{x_r} \\ \frac{z}{f} = \frac{y}{y_l} = \frac{y}{y_r} \end{cases} \tag{1}$$

Then, we can get:

$$\begin{cases} z = \frac{f \times b}{x_l - x_r} = \frac{f \times b}{d} \\ x = \frac{x_l \times z}{f} \\ f = \frac{y_l \times z}{f} \ or \ f = \frac{y_r \times z}{f} \end{cases} \tag{2}$$

In formula (2), $x_l - x_r$ is called parallax $d$, which represents the offset of point $P$ at the projection point corresponding to the left and right camera planes; $Z$ is the depth value of point $P$; According to formula (2), when the parameters $f$ and $b$ are determined, the depth $Z$ of the target point can be obtained only by solving the difference between the $x$ or $y$ coordinates of the target point in the pixel coordinate system of the left and right cameras.

Therefore, in order to obtain the depth information of the target point $P$, it is necessary to calculate the projection point coordinates $p(x_l, y_l)$ and $p(x_r, y_r)$ of the point on the left and right camera imaging planes, and the three-dimensional information of the point can be obtained by converting between coordinate systems.

## 3. Camera Calibration Model

In the BSV ranging technology, in order to obtain the three-dimensional information of the target, the geometric model of the camera imaging must be established, that is, the internal and external parameters of the camera. The process of solving the parameters is called camera calibration [24], and the accuracy and stability of the calibration results will also have a certain impact on the ranging results. The imaging process of the camera adopts the pinhole imaging model [12], the principle of which is portrayed in Figure 3. When shooting with a camera, the light reflected by the object is projected on the imaging plane through the camera lens, indicating that the points in the three-dimensional space are projected on the two-dimensional image through coordinate transformation. This process primarily involves the transformation between four coordinate systems: World Coordinate System, Camera Coordinate System, Image Coordinate System, and Pixel Coordinate System.

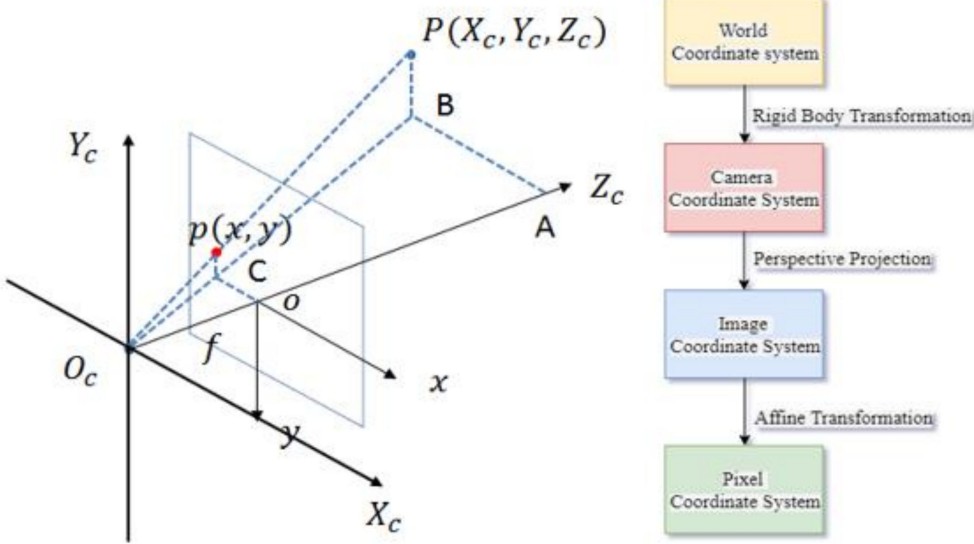

**Figure 3.** Camera imaging model.

According to the principle of pinhole imaging, in Figure 3, assuming a point $P_w(X_w, Y_w, Z_w)$ in the three-dimensional world coordinate system, its projection coordinate in the camera coordinate system is $P_c(X_c, Y_c, Z_c)$, the coordinate in the image coordinate system is $P(x, y)$, and the coordinate in the pixel coordinate system is $P(u, v)$, $O_c$ is the position of the optical center of the cameras, and $O_c Z_c$ is the optical axis of the cameras. Then the relationship between $P_w(X_w, Y_w, Z_w)$ and the pixel coordinate $P(u, v)$ is:

$$
Z_c \begin{pmatrix} u \\ v \\ 1 \end{pmatrix} = \begin{pmatrix} \frac{1}{dx} & 0 & u_0 \\ 0 & \frac{1}{dy} & v_0 \\ 0 & 0 & 1 \end{pmatrix} \begin{pmatrix} f & 0 & 0 & 0 \\ 0 & f & 0 & 0 \\ 0 & 0 & f & 0 \end{pmatrix} \begin{pmatrix} R & T \\ \vec{0} & 1 \end{pmatrix} \begin{pmatrix} X_w \\ Y_w \\ Z_w \\ 1 \end{pmatrix}
$$

$$
= \begin{pmatrix} f_x & 0 & u_0 & 0 \\ 0 & f_y & v_0 & 0 \\ 0 & 0 & 1 & 0 \end{pmatrix} \begin{pmatrix} R & T \\ \vec{0} & 1 \end{pmatrix} \begin{pmatrix} X_w \\ Y_w \\ Z_w \\ 1 \end{pmatrix}
\tag{3}
$$

Then $K = \begin{pmatrix} f_x & 0 & u_0 & 0 \\ 0 & f_y & v_0 & 0 \\ 0 & 0 & 1 & 0 \end{pmatrix}$, which is called the internal parameter matrix of the cameras; the above formula can be simplified to:

$$
Z_c \begin{pmatrix} u \\ v \\ 1 \end{pmatrix} = K \begin{pmatrix} R & T \\ \vec{0} & 1 \end{pmatrix} \begin{pmatrix} X_w \\ Y_w \\ Z_w \\ 1 \end{pmatrix}
\tag{4}
$$

In formulas (3) and (4), $f_x = \frac{f}{d_x}$, $f_y = \frac{f}{d_y}$, $d_x$, and $d_y$ respectively represent the physical size of each pixel in the $x$ and $y$ directions of the image plane, $u_0$ and $v_0$ represent the coordinates of the image center point, $R$ is the rotation matrix, and $T$ is the translation vector; the two constitute the external parameter matrix $\begin{pmatrix} R & T \\ \vec{0} & 1 \end{pmatrix}$ of the camera, and the external parameter and internal parameter matrix of the camera could be obtained by the camera calibration.

## 4. MobileNetV1-YOLOv4 Model

### 4.1. YOLOv4 Model

YOLO (you only look once) is a one-stage target detection algorithm proposed by Redmon et al. in 2016 [25]. This algorithm converts the classification problem of traditional target detection into a regression problem, then the position and probability of the target can be predicted through the input image, and the end-to-end target detection algorithm is realized. This paper builds a ship target recognition model based on the YOLOv4 [26] model, whose network structure is mainly composed of backbone, neck, and head, as portrayed in Figure 4.

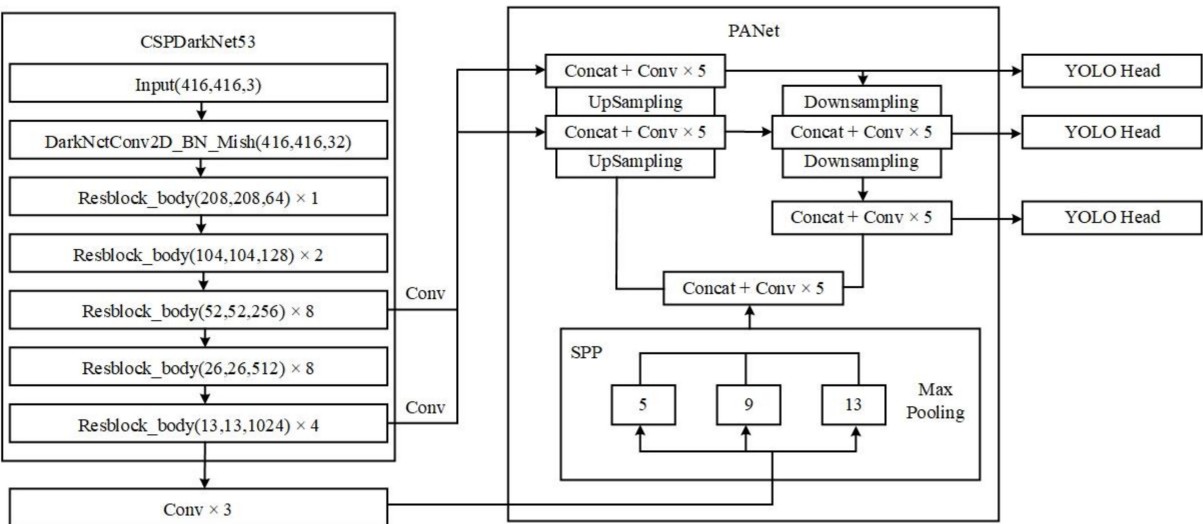

**Figure 4.** YOLOv4 network structure.

The YOLOv4 network adopts CSPDarknet53 as the backbone network for feature extraction. Additionally, it combines the cross stage partial network (CSP) [27] and residual network [28], which effectively solves the problem of duplication of gradient information in the Darknet53 network, and can extract higher-level feature information while reducing the model parameters, the feature extraction capability of backbone network is further enhanced.

The neck module is primarily composed of Spatial Pyramid (SPP) [29] and Path Aggregation Network (PANet) [30]. SPP adopts three pooling layers of different scales: $5 \times 5$, $9 \times 9$, and $13 \times 13$. After max-pooled of the input features, the extent of the receptive field can be greatly increased, thereby eliminating the effects of inconsistencies in the scale of the input image and producing a fixed-length output. The PANet network is used as a feature fusion module, which adds a bottom-up path based on the Feature Pyramid Network (FPN) to improve the model's ability to extract features at different levels. Finally, the head module is used to detect the target and output feature maps of three different sizes: $13 \times 13$, $26 \times 26$, and $52 \times 52$.

### 4.2. MobileNetV1 Model

MobileNet [31] is an efficient and lightweight network specially proposed for mobile and embedded devices. Based on the depthwise separable convolution method, it will decompose the standard convolution into depth wise convolution and the point convolution with the convolution kernel size of $1 \times 1$, the depth convolution convolves each channel of the input image, while the point convolution is used to combine the channel convolution output. This convolution method can effectively reduce the amount of computation and reduce the scale of the model. The flow of standard convolution and depthwise separable convolution is portrayed in Figure 5.

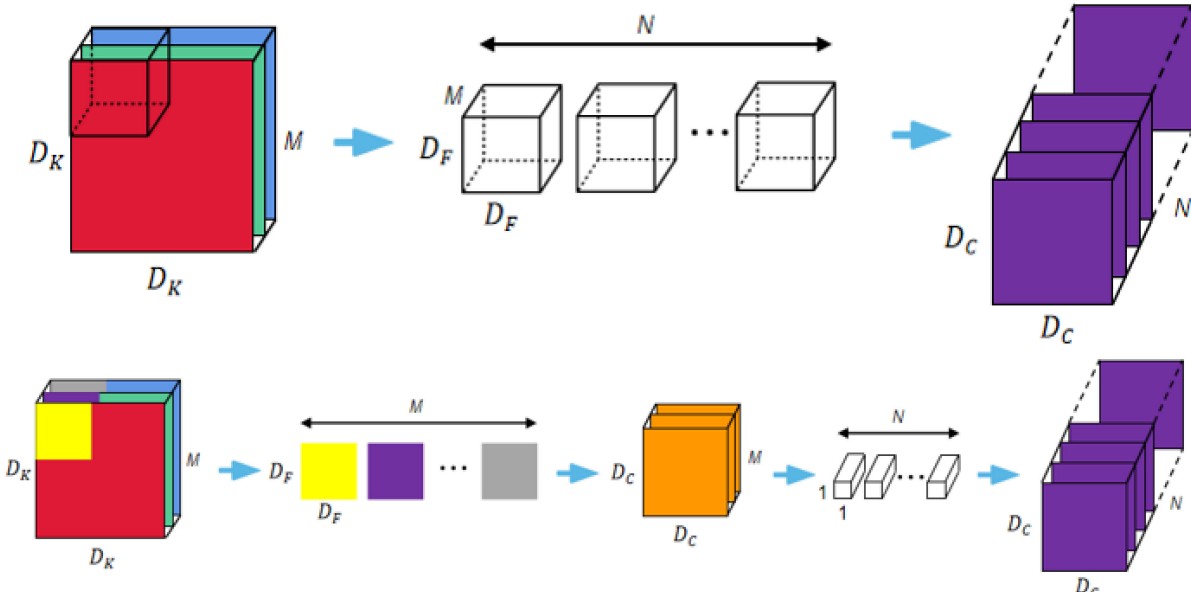

**Figure 5.** Standard convolution and depthwise separable convolution.

Suppose that the size of the input image is $D_k \times D_k$, the input channel is $M$, the number of convolution kernels is $N$, and the size of the convolution kernel is $D_F \times D_F$, then the output feature map size is $D_c \times D_c$, and the output channel is $N$.

In the standard convolution operation, the number of parameters to be calculated is:

$$params1 = D_F \times D_F \times M \times N \tag{5}$$

The calculation of the parameters of the depthwise separable convolution is divided into two parts; one part is the depth convolution parameter, and the other part is the point convolution parameter, where the size of the point convolution kernel is $1 \times 1$. Therefore, the number of parameters that need to be calculated is:

$$params2 = D_F \times D_F \times M + M \times N \tag{6}$$

Then, the ratio of the two is:

$$\frac{params2}{params1} = \frac{D_F \times D_F \times M + M \times N}{D_F \times D_F \times M \times N} = \frac{1}{N} + \frac{1}{D_F^2} \tag{7}$$

According to Equation (7), the number of network parameters can be reduced to a certain extent by setting convolution kernels with different sizes. The MobileNetv1 network is based on the depth-wise separable convolution module, as indicated in Figure 6. A convolution kernel of size $3 \times 3$ is used to perform depth-wise convolution and extract feature information. A BN layer and a ReLU layer are connected between the depth-wise convolution and the point convolution. After the point convolution, the feature information is output through a BN layer and a ReLU layer, and the number of parameters is relatively reduced by eight to nine times, and the convolution effect is equivalent to that of the standard convolution. Reference [31] presents the basic architecture of the MobileNetv1 network, which has 28 layers. The first layer is standard convolution with the size of $3 \times 3$, then 13 depthwise separable convolution modules are built. Before the fully connected layer, an average pooling layer is used to reduce the spatial resolution to 1, and finally the softmax layer is used to output the probability of each class that needs to be identified.

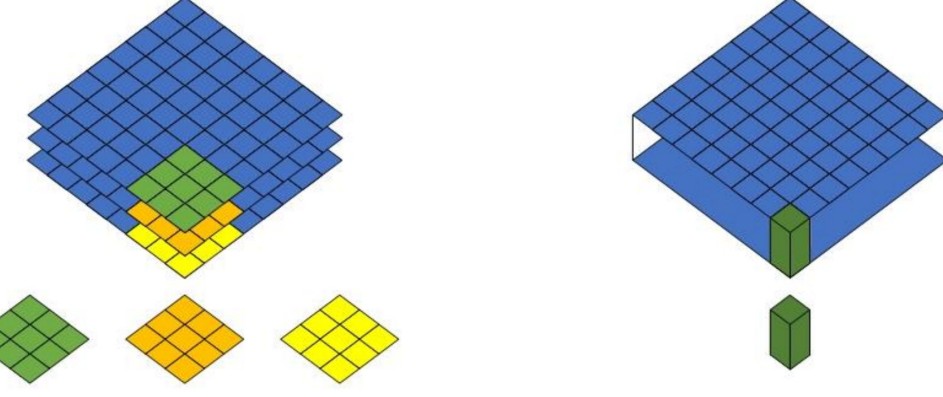

Depthwise Convolutional Filters          Pointwise Convolutional Filters

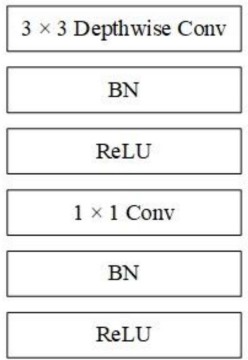

Depthwise Separable Convolution

**Figure 6.** Depthwise separable convolution module.

### 4.3. MobileNetV1-YOLOv4 Model Structure

Considering that the classical YOLOv4 network structure is too large, and the number of parameters calculated is also very large, the trained network model has a large scale and is not suitable for target detection on devices with insufficient computing power and memory. In this paper, the backbone network Darknet53 is replaced by MobileNetV1; that is, the three different scale feature layers extracted by MobileNetv1 are connected directly with the SPP and PANet modules of the classical YOLOv4 model to build the MobilenetV1-YOLOv4 target detection model, which greatly reduces the number of parameters and computation of the model. The MobilenetV1-YOLOv4 network structure is portrayed in Figure 7.

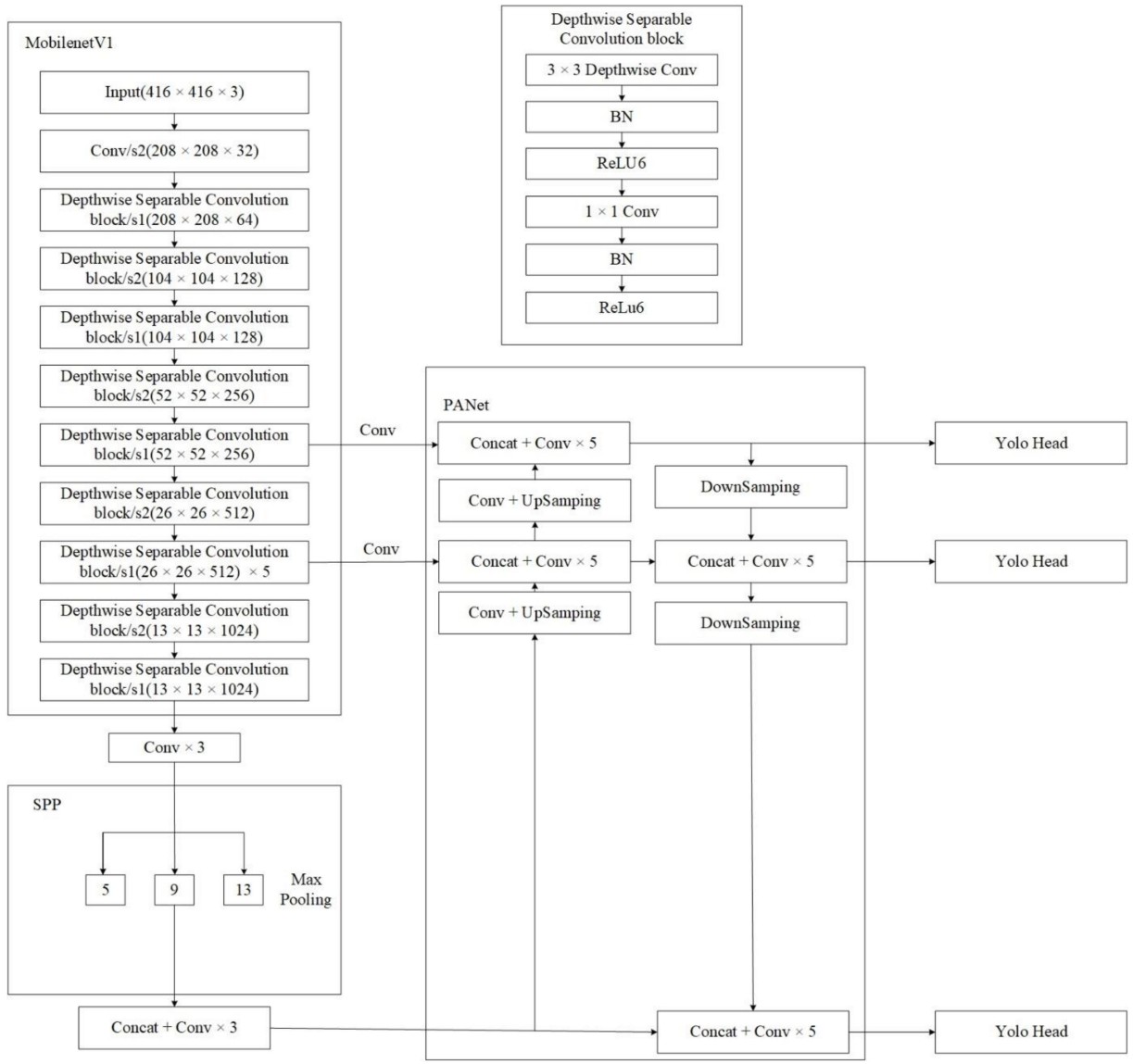

**Figure 7.** Mobilenetv1-YOLOv4 model structure.

## 5. Ship Feature Detection and Matching

### 5.1. FSRCNN Network

The FSRCNN [32] network model consists of five parts: feature extraction, reduction, mapping, expanding, and deconvolution, of which the first four parts are the convolution layer, and the last part is the deconvolution layer. The specific operation process of the network is portrayed in Figure 8.

The FSRCNN network directly uses the original low-resolution image as input and uses a parametric rectified linear unit (PReLU) as the activation function. PReLU is portrayed in Equation (8). The FSRCNN network firstly adopts $d$ convolution kernels of size $3 \times 3$ for feature extraction, then uses $s$ convolution kernels of size $1 \times 1$ to shrink the extracted features, then uses $m$ convolutions of size $3 \times 3$ for concatenation as the mapping layer, and $d$ convolution kernel of size $1 \times 1$ is used for expansion, and at the end of the network, a convolution kernel of size $9 \times 9$ is used for deconvolution to obtain a high-resolution image.

$$f(x_i) = \begin{cases} x_i, x_i > 0 \\ a_i x_i, x_i \leq 0 \end{cases} \tag{8}$$

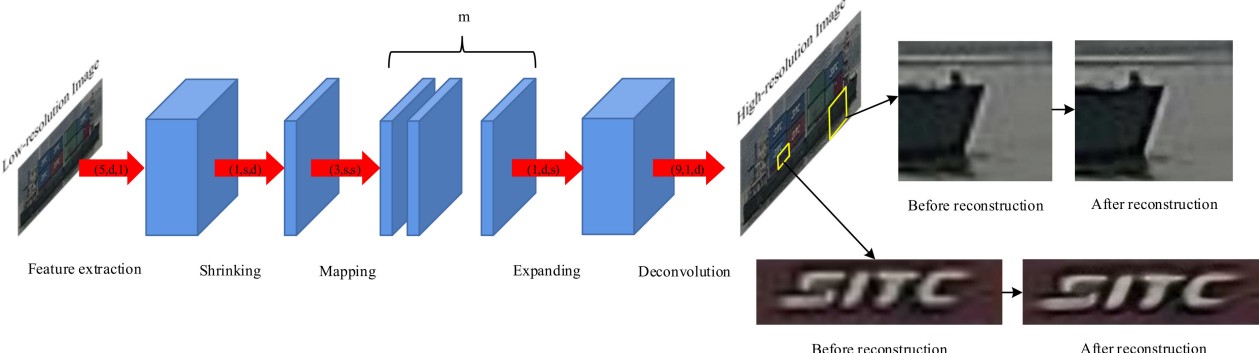

**Figure 8.** FSRCNN model operation.

The mean square error is adopted as the loss function during training of the FSRCNN network, as portrayed in Equation (9):

$$loss = \min_{\theta} \frac{1}{n} \sum_{i=1}^{n} \left|\left| F(Y_s^i; \theta) - X^i \right|\right|_2^2 \qquad (9)$$

where, $Y_s^i$ and $X^i$ are the *i*-th pair of super-resolution images and low-resolution images in the training data, respectively. $F(Y_s^i; \theta)$ is the network output, and $\theta$ is the hyperparameter of the network.

### 5.2. ORB Algorithm

The ORB (Oriented Fast and Rotated BRIEF) [33] algorithm could be used to quickly create feature vectors for key feature points of the image, thereby identifying the corresponding target in the image. Its primary feature is that the detection speed is fast, and it is not restricted by noise and image rotation transformation. The algorithm is primarily divided into three steps:

5.2.1. Feature Points Extraction

The ORB algorithm firstly adopts the FAST (Features from Accelerated Segment Test) [34] algorithm to find the significant feature points in the image. Its primary idea is that if a pixel in the image differs greatly from enough pixels in its neighborhood area, the pixel may be feature points. The specific operations of the algorithm are as follows:

Select a certain pixel point *P* in the image to be detected, as portrayed in Figure 9. The pixel value of the pixel point is $I_P$, and then a circle is determined with *P* as the center and a radius of 3. At this time, there are 16 pixels on the determined circle, which are respectively expressed as: $P_1, P_2, P_3, \ldots, P_{16}$.

a. Determine a threshold: *t*
b. Calculate the difference between all pixel values on the determined circle and the pixel value of point *P*. If there are *N* consecutive points that satisfy Equation (10), then this point could be taken as a candidate point, where $I_x$ represents a certain point of 16 pixels on the circle, according to experience, generally set $N = 12$. Generally, in order to reduce the amount of calculation and speed up the efficiency of feature points search, the pixel points 1, 9, 5, and 13 are detected for each pixel point. If at least three of the four points satisfy the Formula (10), then the point is a candidate detection point.

$$\begin{cases} I_x - I_p > t \\ I_x - I_p < -t \end{cases} \qquad (10)$$

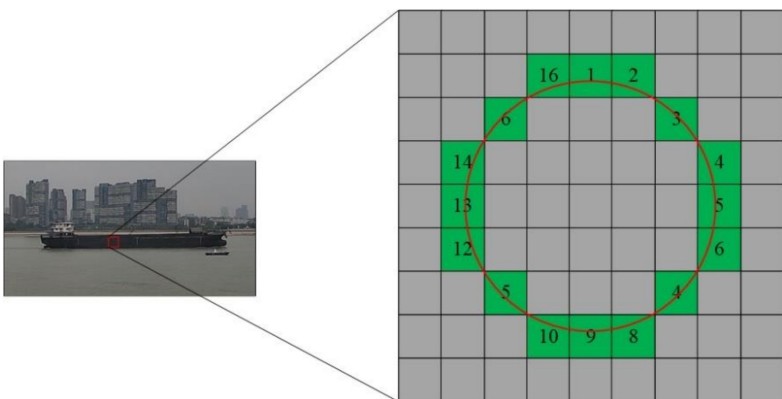

**Figure 9.** FAST feature point detection.

After candidate point detection, multiple feature points are generally detected, and these feature points are likely to be adjacent to each other. To solve this problem, maximum value suppression can be used to delete redundant candidate points.

5.2.2. Build BRIEF Feature Descriptors

The ORB algorithm adopts BRIEF (binary robust independent elementary features) [35] to create binary descriptors for the detected key feature points, whose description feature vectors only contain 0 and 1, thereby speeding up the establishment of feature descriptors. The specific steps are as follows:

a. In order to further reduce the sensitivity of feature points to noise, Gaussian filtering is first performed on the detected image.

b. BRIEF takes the candidate feature point $P$ as the center point, selects a region with size $S \times S$, randomly selects two points $P_x$ and $P_y$ in this region, then compares the pixel sizes of the two points, and performs the following assignments:

$$\tau(p; x, y) = \begin{cases} 1, & p_x < p_y \\ 0, & p_x \geq p_y \end{cases} \tag{11}$$

where, $p_x$ and $p_y$ are the pixel values of random points $x(u_1, v_1)$ and $y(u_2, v_2)$ in the region, respectively.

c. Randomly select $n$ pixel pairs in the region $S \times S$, and binary assignment is performed by the formula (12). This encoding process is the description of the feature points in the image, that is, the feature descriptor. The value of $n$ is usually 128, 258, or 512. while the image features can be described by $n$-bit binary vectors, namely:

$$f_n(p) := \sum_{1 \leq i \leq n} 2^{i-1} \tau(p; x_i, y_i) \tag{12}$$

The ORB algorithm has the characteristic of rotation invariance, and adopts the main direction of the key feature points to rotate the Brief descriptor.

5.2.3. Match the Feature Points

Calculate the Hamming distance between the feature descriptors in the image pair; that is, calculate the similarity between the feature points. If it is less than the given threshold, the two feature points are matched.

### 6. Experiments and Analysis

*6.1. Experiments Environment and Equipment*

The experiment was performed on the Windows 10 system, with i7-11800H 2.30 GHz processor, the GPU was NVIDIA GeForce RTX3060Ti, and the experimental software used Matlab2021b, pycharm2018.3.7, TensorFlow deep learning framework, and OpenCV library.

Among them, Matlab2021b is mainly used to calibrate the binocular camera, and then obtain the corresponding internal and external parameter matrices. And others are used for target recognition, feature point detection, and matching and ranging tasks.

The experimental site is the Nan'anzui Park in Wuhan City, Hubei Province, as indicated in Figure 10a. The ship image data was collected by the Huaxia industrial gun-type camera, and the camera parameters are portrayed in Table 1. The horizontal rotation and pitch angle could be adjusted through the camera bottom holder, and the adjustable range of horizontal distance between the binocular cameras is [0.3 m, 1.5 m], as indicated in Figure 10b.

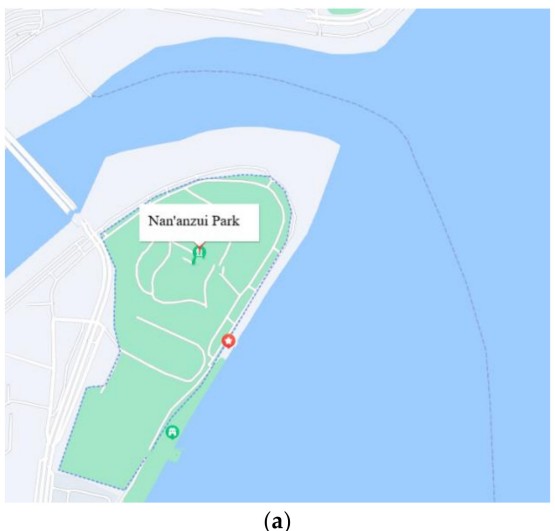

(**a**)

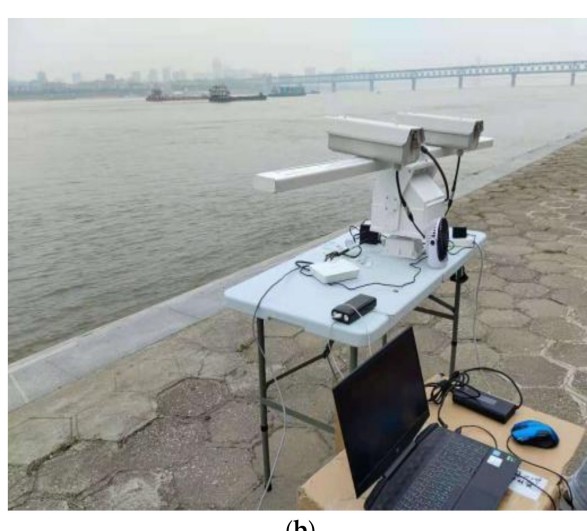

(**b**)

**Figure 10.** Experiments site (**a**) and equipment (**b**).

**Table 1.** Camera Parameters.

| Parameter | Information |
| --- | --- |
| Sensor type | 1/2.8″ Progressive Scan CMOS |
| Electronic shutter | DC Drive |
| Focal length | 5.5–180mm |
| Aperture | F1.5–F4.0 |
| Horizontal field of view | 2.3–60.5° |
| Video compression standard | H.265/H.264/MJPEG |
| Main stream resolution | 50 HZ:25 fps (1920 × 1080, 1280 × 960, 1280 × 720) |
| Interface type | NIC interface |

*6.2. Camera Calibration Analysis*

Camera calibration is the basic work of BSV ranging technology, and the precision of the calibration results is closely related to the ranging effect. In the process of equipment construction, the structural parameters of the camera, the environment and other factors have a certain impact on the calibration results [36]. In the experiment, the horizontal distance between the cameras was set to 50 cm, and the camera was kept level with the river surface. In order to increase the flexibility and operability of the experiment, this paper adopts the calibration method of Zhang [37] for camera calibration experiments. This calibration method is integrated into the MATLAB toolbox Stereo Camera Calibration and is combined with the internal parameter model and distortion model proposed by Heikkil and Silven [38]. It is a commonly used camera calibration method with high precision at present.

In the experiment, a black-and-white checkerboard calibration board of size 16 × 9 was first made, with each grid was 60 mm. The calibration board was photographed from different angles with a binocular camera, 30 sets of calibration board images were taken,

and 18 sets of images were selected after screening. The calibration results are portrayed in Table 2.

**Table 2.** Camera calibration results.

| Parameter | Left Camera | Right Camera |
|---|---|---|
| Internal parameter matrix | $\begin{pmatrix} 1292 & 0 & 611 \\ 0 & 1294 & 375 \\ 0 & 0 & 1 \end{pmatrix}$ | $\begin{pmatrix} 1286 & 0 & 643 \\ 0 & 1289 & 368 \\ 0 & 0 & 1 \end{pmatrix}$ |
| Extrinsic parameter matrix | $R = \begin{pmatrix} 1 & 0.0161 & -0.0033 \\ -0.0161 & 1 & 0.0109 \\ 0.0033 & -0.0109 & 1 \end{pmatrix}$ | $T = [-498 \; 3.949 \; -2.898]$ |
| Distortion coefficient matrix | $[-0.299 \; 0.164 \; 0 \; 0 \; 0]$ | $[-0.281 \; 0.079 \; 0 \; 0 \; 0]$ |

In Table 2, it can be observed from the internal parameter matrix that the focal length of camera is approximately the same, and the rotation matrix is similar to the identity matrix, indicating that the two cameras are essentially in a parallel state. The reprojection error of the camera is indicated in Figure 11. The maximum calibration error is 0.40 pixels, and the average error is 0.24 pixels, both of which are less than 1 pixel, reaching the standard for experimental use [36]. The obtained parameters can be used for stereo correction processing of the images.

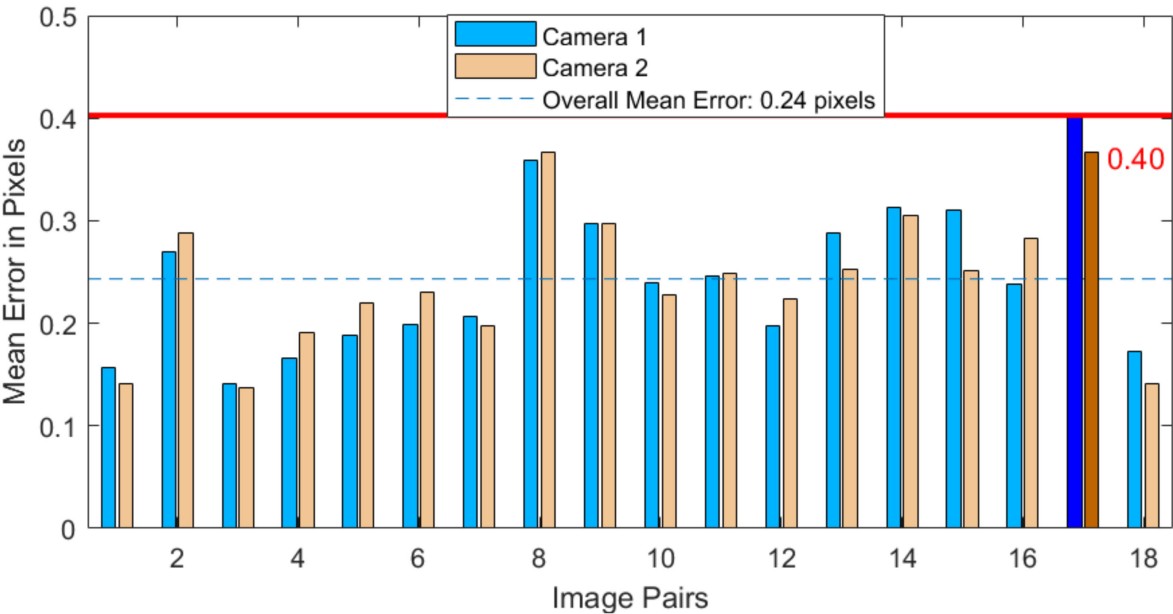

**Figure 11.** Reprojection error of camera calibration.

*6.3. Ship Target Recognition Analysis*

6.3.1. Ship Images Collection and Labeling

In the experiment, a total of 2000 images of inland river ships were collected by binocular stereo camera with a resolution of 1280 × 720, including container ship, passenger ship, and ore carrier. Before the ship image is input into MobileV1-YOLOv4 network, certain annotation operations need to be performed on the image. In this paper, the LabelImg tool is used to label the ship target, and the annotation results include the position information of the ship target box, that is, the coordinate value of the target, category information, etc. Part of the ship image data and the annotation process are portrayed in Figure 12.

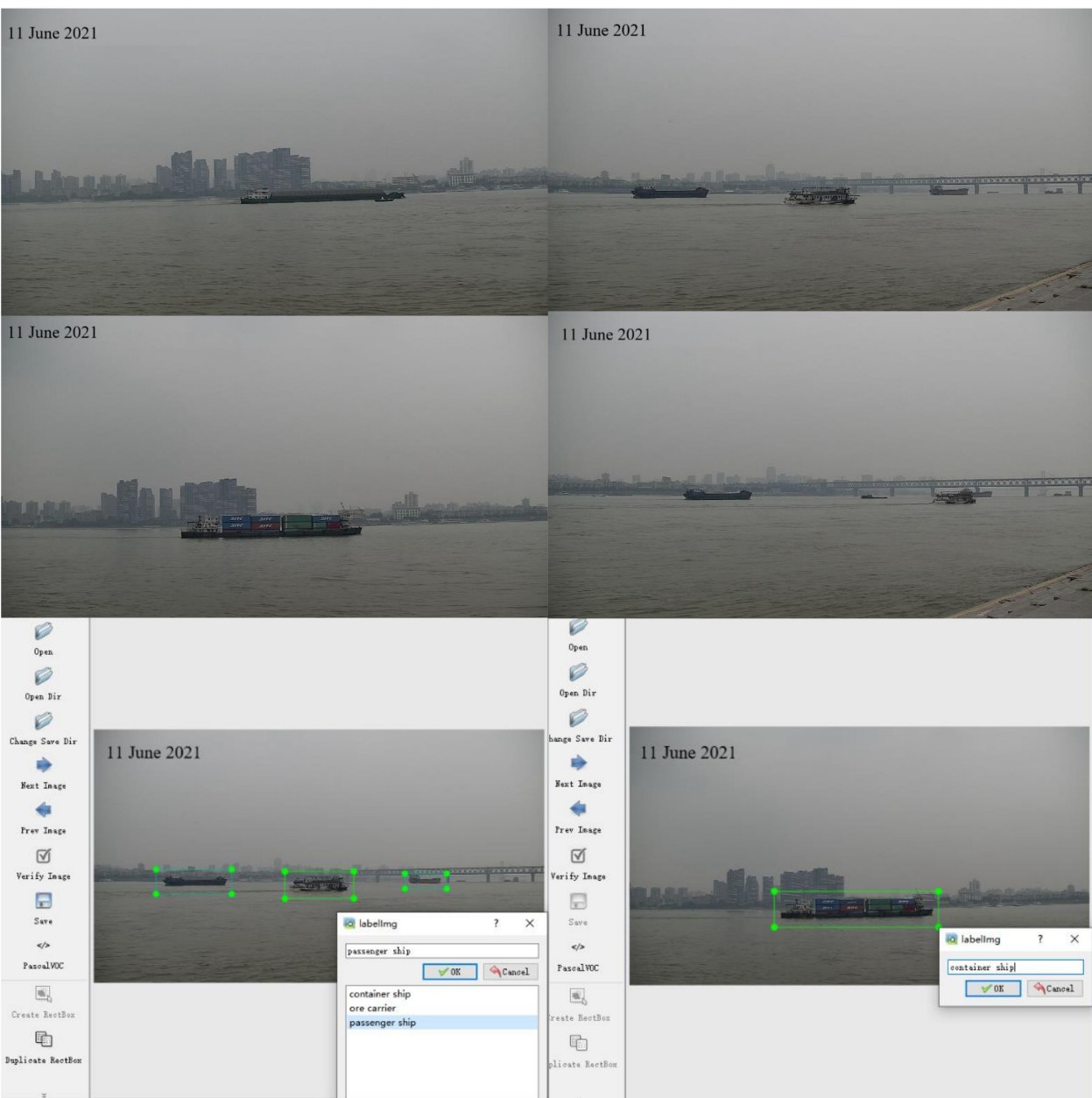

**Figure 12.** Ship image and annotation process.

### 6.3.2. Model Evaluation Index

The evaluation index [26] of target detection in deep learning is the basis to measure the quality of detection results. In this paper, the precision of the detection results (precision), the recall rate (recall), the category evaluation accuracy rate mAP (mean average precision) and F1-score are comprehensively considered. The mathematical expressions of each index are portrayed in Equations (13)–(16):

$$Precision = \frac{TP}{TP + FP} \tag{13}$$

$$Recall = \frac{TP}{TP + FN} \tag{14}$$

$$mAP = \frac{1}{classes} \sum_{i=1}^{classes} \int_{0}^{1} P(R)d(R) \tag{15}$$

$$F1 = \frac{2}{\frac{1}{Precision} + \frac{1}{Recall}} = 2 \cdot \frac{Precision \cdot Recall}{Precision + Recall} \qquad (16)$$

where $TP$ (True Positives) represents the number of positive samples detected as positive samples; $FN$ (False Negatives) represents the number of positive samples detected as negative samples; $FP$ (False Positives) represents the number of negative samples detected as positive samples, $mAP$ represents the average area under the curve of multiple samples, which is used as a measure of detection accuracy in target detection. *classes* is the detection category, where $classes = 3$, $F1$ represents the summed average of precision and recall.

### 6.3.3. Ship Target Recognition

In the part of target recognition, the K-means algorithm is firstly adopted to obtain the prior anchor boxes, and each scale generates three anchor boxes with different sizes, therefore 9 anchor boxes are generated here, and the size of each anchor box is provided in Table 3.

**Table 3.** The anchor boxes of MobileV1-YOLOv4.

| (17, 6) | (18, 8) | (19, 12) |
|---|---|---|
| (28, 10) | (29, 14) | (51, 9) |
| (44, 16) | (71, 19) | (118, 27) |

In this paper, considering comprehensively the shooting target scenes of diversity, firstly, the Mosaic technology is adopted to preprocess the ship images through randomly cropping and stitching the collected ship images from multiple angles, which further enriches the data samples. In the experiment, the data set is randomly divided into training set and test set according to the ratio of 9:1. In order to verify the effectiveness of the improved YOLOv4 algorithm, the classic YOLOv4 algorithm is used for comparative experiments. Considering the influence of the hyperparameters of the neural network on the overall performance of the model, according to the previous experience, this paper continuously fine-tunes the hyperparameters of the model, and after many experiments, the key parameters of the model are set as follows: $learning\_rate = 0.001$, $batch\_size = 4$, $optimizer = SGD$, $epoch = 300$. The loss curve trained by the MobileV1-YOLOv4 network is portrayed in Figure 13. It can be observed that after 300 epochs of training, about 4.8 h, the loss value decreases continuously and finally converged to 0.1, and a better training model is obtained.

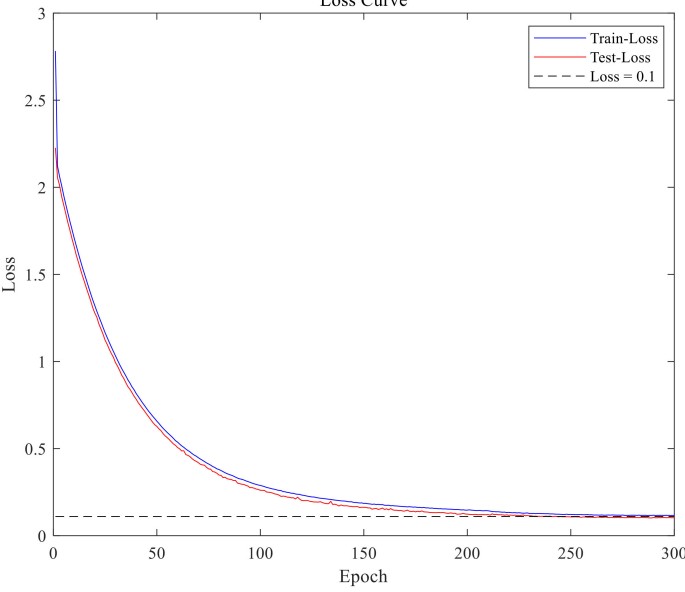

**Figure 13.** The loss curve trained of MobileV1-YOLOv4 model.



The test set data is input into the trained model, and the model recognition results are portrayed in Figure 14. The Mobilv1-yolov4 model can accurately recognize different types of ship targets with high accuracy.

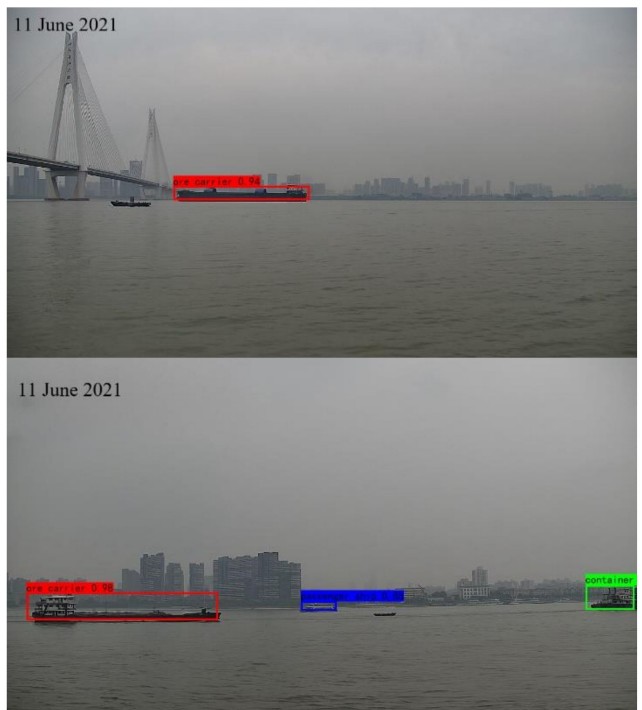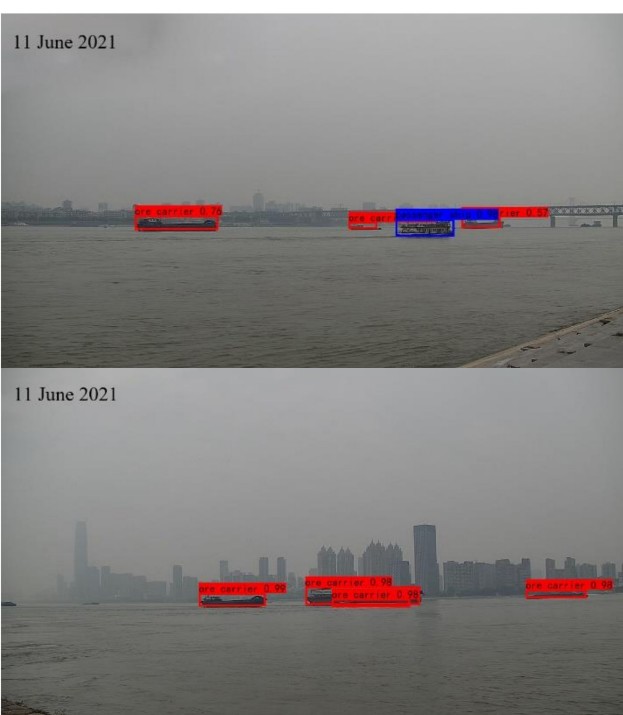

**Figure 14.** The ship recognition of MobileV1-YOLOv4 model.

In the quantitative evaluation stage of model performance, in order to better compare the advantages and disadvantages of the two models, both are performed on the Python platform, and the key parameters of the model are set as: $learning\_rate = 0.001$, $batch\_size = 4$, $optimizer = SGD$, $epoch = 300$. The six indicators including Precision, Recall, mAP, FPS, and Backbone_weights are used for evaluation and analysis. The experimental comparison results of the two models are portrayed in Table 4.

**Table 4.** Comparison of different models.

| Model | Classes | Input_size | Score_threshold | Precision/% | Recall/% | mAP/% | Backbone_weight/M | FPS | F1-Score |
|---|---|---|---|---|---|---|---|---|---|
| YOLOv4 | Ore carrier | $416 \times 416$ | 0.5 | 91.23 | 86.21 | 90.70 | 244 | 26.11 | 0.89 |
| | Container ship | | | 100.00 | 100.00 | | | | 1 |
| | Passenger ship | | | 87.10 | 75.00 | | | | 0.81 |
| MobilevV1-YOLOv4 | Ore carrier | $416 \times 416$ | 0.5 | 87.95 | 84.91 | 89.25 | 47.6 | 66.23 | 0.86 |
| | Container Ship | | | 100.00 | 100.00 | | | | 1 |
| | Passenger ship | | | 89.29 | 69.44 | | | | 0.78 |

Table 4 portrays the comparative experimental results of the two models. The two models have different advantages and disadvantages under different indicators. Under the same input dimension, the YOLOv4 model and the MobileV1-YOLOv4 model both portray strong advantages in the recognition results of the three types of ship. For the container ship, the two indicators of precision and recall both reach 100%, while the recognition effect of ore carrier and passenger ship is slightly lower. This is because there is more container ship in the scene, which is more conducive to model training, and the ore carrier and passenger ship are fewer, which affects the recognition results to a certain extent. Generally, the mAP

value of both reaches about 90%. F1-score is a comprehensive evaluation index to measure precision and recall; it is evident from the experimental results that the F1-score values of the two models are both high, that is, both precision and recall have reached a balance. For the weight size of the backbone network, the weight of the MobileV1-YOLOv4 model is only about 20% of that of the YOLOv4 model, which greatly relieves the computing and memory pressure of the computer, and the FPS of the MobileV1-YOLOv4 model reaches 66.23, while that of Yolov4 model is only 26.11. Replacing the feature extraction module of the classic YOLOv4 network with the MobileV1 network greatly reduces the network parameters, improves the calculation speed, and has high real-time performance on the basis of ensuring high recognition accuracy.

### 6.4. Ship Target Depth Estimation Analysis
6.4.1. Ship Features Detection and Matching

In practical applications, due to factors such as the environment and camera, when using the matching algorithm based on grayscale correlation, the feature points in the image are relatively sparse, which is not easy to detect and recognize with respect to feature points, and it is difficult to obtain accurate image matching results. Thereby it is difficult to perform parallax and will affect the depth estimation results. To solve these problems, this paper proposes a sub-pixel feature point matching method based on region. Firstly, the ship region is extracted through the ship target bounding box output by the mobilev1-yolov4 network, and then the FSRCNN network is adopted to super-resolution enhancement of the ship region, and further increase the number of feature points, and finally the ORB algorithm is used to detect and match ship feature points.

In this paper, the general data set ImageNet-91 is used to train the FSRCNN network, and the data set Set14 made by us was used as the test set. In the training process of FSRCNN network, the key parameters of the model are set as: $learning\_rate = 1 \times 10^{-5}$, $optimizer = Adam$, $epoch = 60K$. The result after reconstruction by the FSRCNN network is portrayed in Figure 15. After the super-resolution reconstruction of the ship image, the edge feature information is enhanced, which makes the image clearer and achieves the effect of denoising to a certain extent.

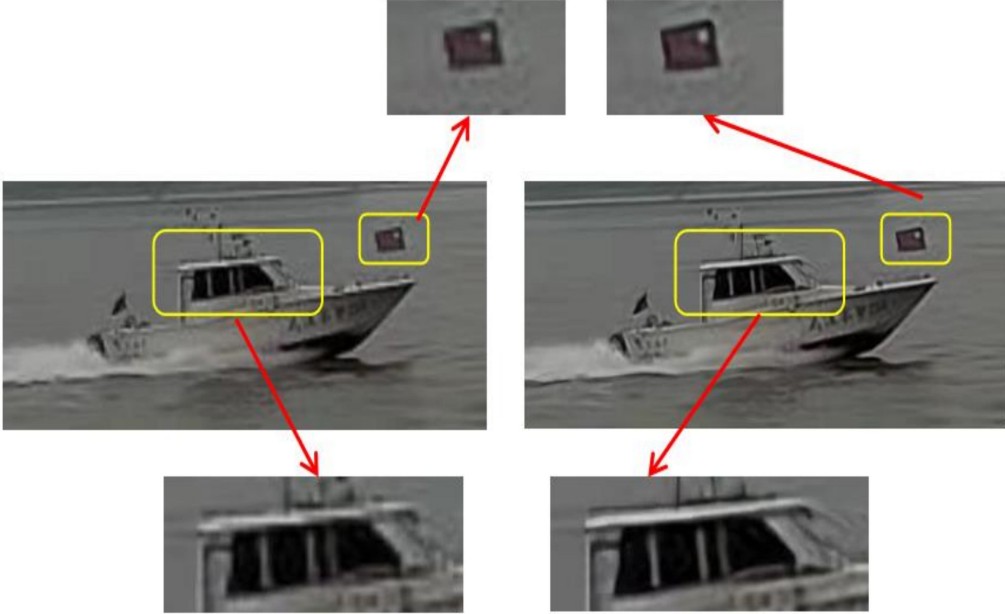

**Figure 15.** The result of ship by super-resolution reconstruction.

In order to further quantify the performance of the FSRCNN network, the peak signal to noise ratio (PSNR) [32] was adopted for evaluation, and a comparative experiment is conducted with the ESPCN [39] method. The experimental results of some pictures in the

test set are portrayed in Table 5. It can be observed from Table 5 that the PSNR of the image reconstructed by the FSRCNN method is higher, which means that the image quality is better and the distortion is smaller; for images of different sizes, the reconstruction time of the two methods is equivalent, and the reconstruction time of both methods shorter is generally.

**Table 5.** The reconstruction results of test set images.

| Test_picture | Picture_size | Model | PSNR/dB | Time/s |
|---|---|---|---|---|
| Test_pic1 | 456 × 72 | FSRCNN | 35.945062 | 0.045590 |
| | | ESPCNN | 34.582875 | 0.027407 |
| Test_pic2 | 381 × 74 | FSRCNN | 35.562458 | 0.018695 |
| | | ESPCNN | 36.029904 | 0.016069 |
| Test_pic3 | 193 × 43 | FSRCNN | 35.875411 | 0.006879 |
| | | ESPCNN | 35.246397 | 0.007040 |
| Test_pic4 | 426 × 72 | FSRCNN | 38.673282 | 0.019900 |
| | | ESPCNN | 38.022336 | 0.016829 |
| Test_pic5 | 540 × 70 | FSRCNN | 38.444051 | 0.027066 |
| | | ESPCNN | 37.565404 | 0.029988 |
| Test_pic6 | 88 × 211 | FSRCNN | 36.462584 | 0.017341 |
| | | ESPCNN | 34.900440 | 0.012008 |

The ORB algorithm is used for feature point extraction and matching of the ship image after super-resolution reconstruction. The algorithm performs down-sampling feature extraction by constructing an image pyramid, and performs feature point detection on each down sampled image based on the FAST algorithm. The feature point extraction and matching result is portrayed in Figure 16. The ORB algorithm can obtain a better feature point matching result. The matching pair contains a large amount of edge and contour information of the image, and the distribution is more uniform, which is more conducive to the calculation of parallax.

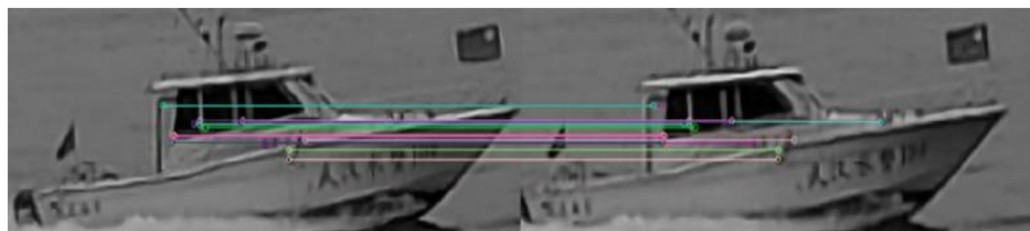

**Figure 16.** The result of ship feature matching by ORB.

6.4.2. Ship Target Depth Estimation

In the experiment, the ORB algorithm will detect multiple feature points and obtain more disparity values. In this paper, the average disparity value of all feature points of a single target is used to calculate the target depth. The results of depth estimation on different type ships are portrayed in Figure 17.

In the experiment, a bulk cargo carrier was taken as an example for continuous depth estimation, as portrayed in Figure 18, where the coordinate (0,0) is the position of the camera. Most of the ships captured by the binocular camera are in the state of direct sailing, and this ship is sailing roughly in a straight line. For the depth estimation of the bulk cargo carrier, the calculated depth of the ship has little fluctuation, and maintains a stable state, which is consistent with the sailing state of the ship.

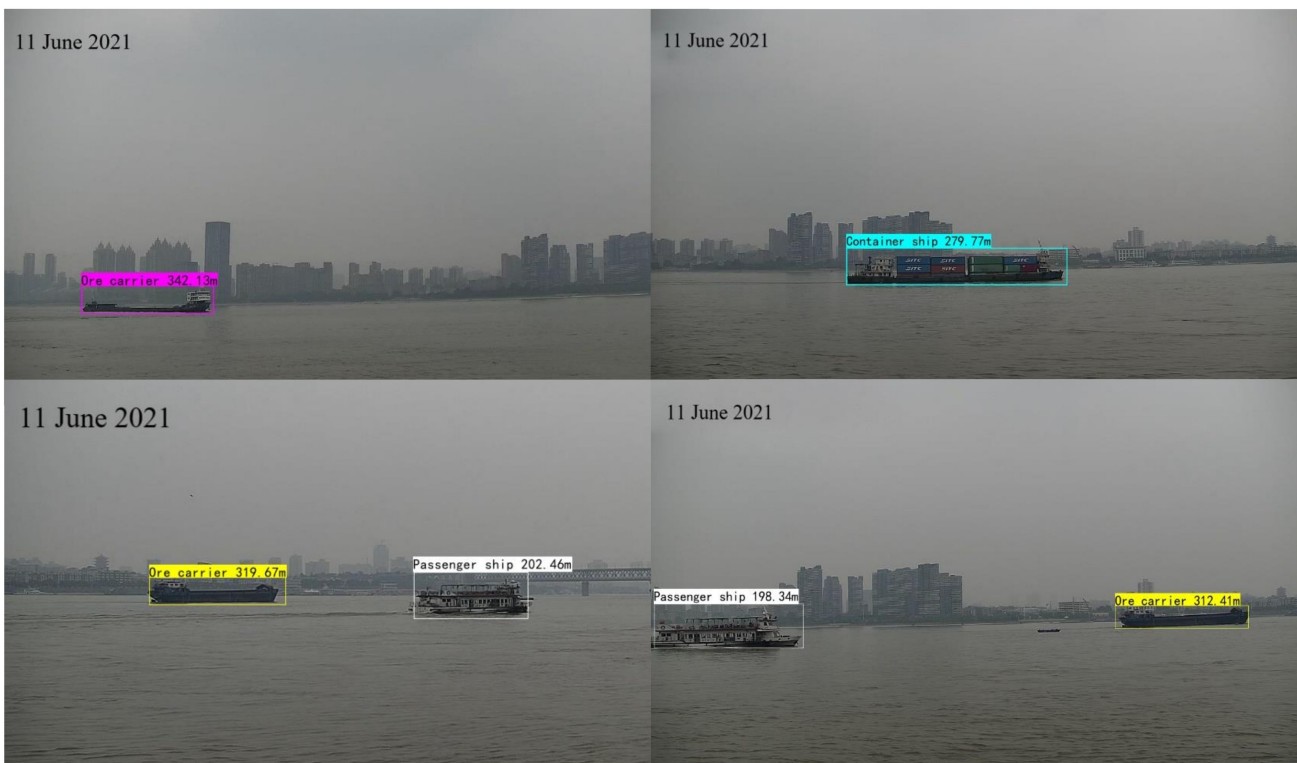

**Figure 17.** The results of depth estimation on different -type ships.

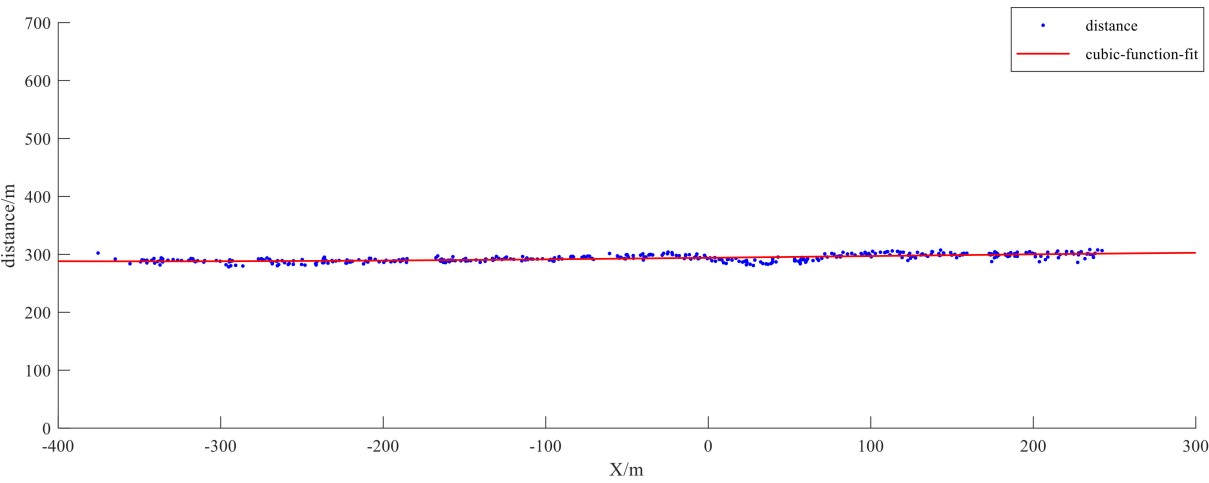

**Figure 18.** The depth estimation of bulk cargo carrier.

In order to further illustrate the effectiveness of the binocular depth estimation experiment, the SNDWAY-1000A laser rangefinder is used for verification in this paper. The distance measurement error of the device within 1000 m is $\pm 0.5$ m, and the error analysis is performed with the distance measured by the rangefinder as the standard. The ranging comparison results of different ships are portrayed in Table 6.

It can be observed from Table 4 that the BSV depth estimation technology proposed in this paper can realize the ranging of ships at different distances. When the laser rangefinder is used as the standard, there is a certain error in the depth calculated by this technology. When the target depth is about 300 m, the error is less than $\pm 2$%. When the target depth is greater than 300 m, the error increases. The increase of the target depth will further increase the difficulty of detection and matching of feature points, which will affect the calculation results of parallax. As a result, the error will increase to a certain extent, but overall, the

error is within ±3%. In the process of navigation in inland river ships, the depth of about 300 m can satisfy the ships to make corresponding decisions. Therefore, the BSV technology can meet the depth estimation requirements of inland ships and has important research significance for the development of future intelligent ships.

**Table 6.** The depth estimation comparison results of different ships.

| Ship_num | BSV Depth Estimation/m | Laser Depth Estimation/m | Depth Estimation Error/m | Error Rate |
|---|---|---|---|---|
| Ship_1 | 105.10 | 103.80 | +1.30 | 1.25% |
| Ship_2 | 122.13 | 124.50 | −2.37 | −1.90% |
| Ship_3 | 168.31 | 166.30 | +2.01 | 1.21% |
| Ship_4 | 198.21 | 195.30 | +2.91 | 1.49% |
| Ship_5 | 220.92 | 224.60 | −3.68 | −1.63% |
| Ship_6 | 245.35 | 248.50 | −3.15 | −1.27% |
| Ship_7 | 279.02 | 275.40 | +3.62 | 1.31% |
| Ship_8 | 285.76 | 290.20 | −4.44 | −1.53% |
| Ship_9 | 311.26 | 305.80 | +5.46 | 1.97% |
| Ship_10 | 348.08 | 355.30 | −7.22 | −2.03% |

## 7. Discussion

With the deepening of economic globalization, water transportation has gradually become one of the most important modes of transportation in international trade. At present, the number, types and new routes of ships are increasing. Although the shipping industry provides a thriving atmosphere, it also makes the channel congested and the load increase, which affects the safety of ship navigation and seriously threaten the life and property safety of ship personnel. From the analysis of the ship accident investigation organization, human error is the primary cause of marine and inland river accidents. The key to the safe navigation of ships lies in the perception of the surrounding navigation environment and the effective use of various perception information for correct analysis and decision-making. As a common navigation environment perception method, AIS has certain limitations in the process of receiving and sending ship information. It also restricts the ship's maneuvering behavior and affects the safe navigation of the ship to a certain extent.

Based on the above reason, this paper applies binocular stereo vision technology to the recognition and depth estimation of inland ships; this technology makes up for the deficiencies of the existing environmental perception methods to a certain extent, however, this technology still has some limitations: firstly, in the stage of ships' recognition, deep learning relies on a large amount of data to continuously train to achieve higher recognition accuracy. The inland river scenes photographed in this paper are relatively simple, and there are fewer types of ships, which indirectly affects the overall recognition performance of the model. Secondly, this paper uses the ORB algorithm to detect and extract ship features points, however, in practical applications, environmental factors such as weather and illumination affect the quality of the collected images, which makes the depth estimation results of ship targets unsatisfactory. In view of these deficiencies, further research will be performed in the next work.

## 8. Conclusions

Aiming at the insufficiency of the existing means of environmental perception in the process of navigation of inland river ships, this paper applies binocular stereo vision technology to the recognition and depth estimation of inland river ships. This work is primarily divided into two stages: ship target recognition and depth estimation. In the stage of ship recognition, based on the classic YOLO-V4 network model, considering the computational pressure brought by the huge network parameters of the model, a lightweight network is proposed to complete the recognition task; that is, the MobileNetV1 network will replace the feature extraction network CSPDarknet53 of the YOLOv4 model,

the experiment result indicates that the mAP value of the MobileNetV1-YOLOv4 model reaches 89.25%, and the weight size of the backbone network is only 20% of that of the classic YOLOv4 network, which greatly reduces the amount of calculation while ensuring the recognition accuracy. In the stage of ships' depth estimation, a feature point detection and matching algorithm based on sub-pixel level is proposed; that is, based on the ORB algorithm, the FSRCNN network is used to perform super-resolution reconstruction of the image pair to further enhance the ship feature information, which is more conducive to calculation of image disparity values. Through the depth estimation experiments on different type ships, when the depth to the target is about 300 m, the depth estimation error is less than 3%, and the depth estimation accuracy is high. The binocular stereo vision technology proposed in this paper further enriches the ship's perception of the navigation environment, improves the safety of inland waterway navigation, and has important research significance for the development of intelligent ships in the future.

**Author Contributions:** Conceptualization, Y.Z. and L.Q.; Data curation, P.L., S.Q., X.L. and G.C.; Formal analysis, L.Q.; Funding acquisition, Y.Z.; Writing-original draft, L.Q.; Writing-review and editing, Y.Z. and Y.M.; Resources, Y.Z.; Validation, P.L.; Methodology, L.Q. and Y.M.; Software, X.L. All authors have read and agreed to the published version of the manuscript.

**Funding:** This research was funded by the National Natural Science Foundation of China (No. 51979215, No.52171350, No.52171349).

**Institutional Review Board Statement:** Not applicable.

**Informed Consent Statement:** Not applicable.

**Data Availability Statement:** The data presented in this study are available on request from the corresponding author.

**Acknowledgments:** The research is financially supported by National Nature Science Foundation of China (51979215; 52171350; 52171349).

**Conflicts of Interest:** The authors declare that they have no known competing financial interests or personal relationships that could have appeared to influence the work reported in this paper.

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
