# Peer review of "Recognition and Depth Estimation of Ships Based on Binocular Stereo Vision"

_jmse, doi:10.3390/jmse10081153_

Round 1

Reviewer 1 Report

Motivation of the study is not very clear. Authors should clearly mention the objectives of the study pointwise in introduction section.

It is suggested that the authors study the f1-score while evaluating the index of target detection.

Picture quality should be improved.

Contributions are required to be written separately and better to clarify in the discussion part.

All references should be in the journal format.

Author Response

Point 1: Motivation of the study is not very clear. Authors should clearly mention the objectives of the study pointwise in introduction section.

Response 1: Thanks to the reviewer for their comments, the revision of point 1 is as follows:

The motivation of this paper is that our country's shipping industry has also made significant progress, and the number of marine, especially inland river ships, is increasing year by year, which not only promotes the national economic development, but also leads to the rising trend of ship traffic accidents. In order to further improve the safety of ship navigation and overcome the shortcomings of the existing navigation environment perception methods, such as AIS, then this paper applies BSV technology to the recognition and depth estimation of inland ships. And in the introduction section of revised manuscript, authors have clearly described the objectives of the study.

Point 2: It is suggested that the authors study the f1-score while evaluating the index of target detection.

Response 2: Thanks to the reviewer for their comments, the revision of point 2 is as follows:

F1-score represents the summed average of Precision and Recall.

Authors have analyzed the F1-score to evaluate the performance of models. The experimental comparison results of the two model are shown in Table 4, and the values of F1 is analyzed in detail.

Point 3: Picture quality should be improved.

Response 3: Thanks to the reviewer for their comments, the revision of point 3 is as follows:

In the revised manuscript, the quality of pictures has been improved.

Point 4: Contributions are required to be written separately and better to clarify in the discussion part.

Response 4: Thanks to the reviewer for their comments, the revision of point 4 is as follows:

In the revised manuscript, refer to articles published in this journal, Discussion and Conclusion are analyzed separately, see the content of the revised manuscript for detail.

Point 5: All references should be in the journal format.

Response 5: Thanks to the reviewer for their comments, the revision of point 5 is as follows:

In the revised manuscript, all references have been in the journal format.

Reviewer 2 Report

This paper presents a study to recognize and estiamte the depth of inland river ships based on binocular stereo vision (BSV) using neural networks. The authors results showed that MoileNetV1-YOLOv4 model greatly reduced the amount of computation while ensuring the recognition accuracy. This is very interesting and useful for navigation safety, thereby aligning with the JMSe journal's scope. I would recommend accept after incorporating these following comments:

1. Please describe why hyperparameter tuning is not performed? is it because of the computational cost?

2. Please provide more details on the computational cost to trained the network for your problem

3. Was there any data augmentation performed to improve the network's performance? If not, why such a approach was not tried? Please explain it in a paragraph?

Author Response

Point 1: Please describe why hyperparameter tuning is not performed? is it because of the computational cost?

Response 1: Thanks to the reviewer for their comments, the revision of point 1 is as follows:

The hyperparameters of neural network are difficult to determine. In this paper, considering model calculation complexity, calculation time, and prediction performance for the model, according to the previous experience, and do some tune to the hyperparameters of neural network, after many comparative experiments, choose the better experimental results as the final model parameters, In the revised manuscript, this paper has also clearly described the reason.

Point 2: Please provide more details on the computational cost to trained the network for your problem.

Response 2: Thanks to the reviewer for their comments, the revision of point 2 is as follows:

Combined with point 1, in order to make the network get better performance, it is necessary to continuously tune the parameters in the network. In point 1, the determination of hyperparameters is based on previous experience and continuous fine-tuning, and each training and parameter tuning requires spend a lot of time. The mobilev1 network is used to replace the backbone module of YOLOv4, which further shortens the training time of the network, about 4.8h. In the experimental analysis part, this paper calculates the size of the backbone of different models, and uses FPS as an indicator of the real-time performance of the network, The FPS of Mobilev1-yolov4 network can reach 66.23, which has high real-time performance. The result is also limited by computer performance. If the experiment is performed on a high-performance computer, it will have higher real-time performance. Moreover, once the established model has a good training result, the obtained weight parameters also have certain generalization performance, which can be fine-tuned or used directly in subsequent experiments. Therefore, this paper mainly reflects the recognition accuracy and real-time. In the revised manuscript, the time required to train the network is also provided, see the experimental analysis section for details.

Point 3: Was there any data augmentation performed to improve the network's performance? If not, why such an approach was not tried? Please explain it in a paragraph?

Response 3: Thanks to the reviewer for their comments, the revision of point 3 is as follows:

In this paper, considering comprehensively the shooting target scenes of diversity, firstly, the Mosaic technology is adopted to preprocess the ship images, that is through randomly cropping and stitching the collected ship images from multiple angles, which further enriches the data samples.

In the revised manuscript, this paper has introduced the data augmentation algorithm, the Mosaic technology uses four pictures, which has the advantage of enriching the background of the detected object, and the data of the four pictures will be calculated at once during the BN calculation, so that the size of mini-batch does not need to be large, then a GPU can achieve a better result.

Round 2

Reviewer 1 Report

The article has been revised based on previous comments. In my opinion, the article is now suitable to publish by the journal.